# Physicochemical and Functional Properties of DND358 (A Hypocholesterolemic Soybean) Protein Isolate

**DOI:** 10.3390/foods13203236

**Published:** 2024-10-11

**Authors:** Tingting Luo, Yuanhang Fan, Mengmeng Fan, Ming Li, Zhendong Qiu, Qiuyan Du, Chongxuan Ma, Chang Liu, Yuhan Peng, Shuzhen Zhang, Shanshan Liu, Bo Song

**Affiliations:** 1Soybean Research Institute, Northeast Agricultural University, Harbin 150030, China; 2Keshan Branch of Heilongjiang Academy of Agricultural Sciences, Qiqihar 161000, China; 3Key Laboratory of Molecular and Cytogenetics, College of Life Sciences and Technology, Harbin Normal University, Harbin 150025, China

**Keywords:** soybean, extraction conditions, functional feature description

## Abstract

The properties and applications of soybean protein isolates (SPIs) have been extensively investigated. In this study, we determined the optimal conditions for the preparation of the DND358 soybean protein isolate (DND358-SPI), assessed its physicochemical and functional properties, and investigated its potential applications in the food industry. According to the results, the highest extraction rate of DND358-SPI was observed when the pH was 9.5, the temperature was 55 °C, the duration was 80 min, and the material-to-liquid ratio was 1:20 (*w*/*v*). With regard to the functional properties, the water-holding capacity (WHC) and oil-binding capacity (OBC) of DND358-SPI were higher than those of other varieties, reaching 4.73% and 11.04%, respectively. In addition, the hardness, adhesiveness, chewiness, and resilience of DND358-SPI were higher than those of other varieties, reaching 159.27 g, 186.07 g, 6.78 mj, and 1.88, respectively. These findings indicate that DND358-SPI can reduce cholesterol levels and may be used to produce cholesterol-lowering food products.

## 1. Introduction

Soybean is a major focus of current research owing to its strong cholesterol-lowering effects [1]. The predominance of soy protein, as well as specific soy peptides, and the content of isoflavones and saponins in soybean are thought to influence the cholesterol-lowering response [2,3]. A soy protein-rich diet has been shown to reduce cholesterol levels in patients with hypercholesterolemia [4]. The cholesterol-lowering effects of soy protein were first reported as early as 1967 in the Asahi Shimbun in Japan [5] and have been investigated in clinical settings for more than 50 years. In 1995, a meta-analysis of 31 trials, involving 564 participants, reported that soy protein resulted in an estimated 12.9% decrease in low-density lipoprotein cholesterol (LDL-C) levels [6]. In addition, we have previously demonstrated that 7S globulin has excellent cholesterol-lowering effects and have identified a large number of α’ subunits. High doses of the α’ subunit of 7S macroglobulin can reduce the plasma cholesterol and triglyceride levels in rats with hypercholesterolemia, with similar efficacy to a 10-fold amount of clofibrate [7,8,9].

Soy protein isolate (SPI) is a complete protein with an ideal amino acid profile, comprising essential amino acids, such as lysine and methionine, and non-essential amino acids, such as arginine and glutamine. It is the purest form of protein in soybean, with the minimum protein content of 90% (dry weight), and is obtained by extracting soluble proteins and removing non-protein components [10]. SPI is prepared by dissolving soymeal-derived protein at a high pH, collecting the supernatant via centrifugation, and reprecipitating the protein in the supernatant at the isoelectric point [11,12,13]. To date, several methods have been used to isolate proteins from plants [14,15]. The most common methods of extracting SPI primarily include alkaline solubilization with acid precipitation, ethanol extraction, fermentation, enzymatic extraction, and ion exchange membrane-based separation [16]. Alkaline solubilization and acid precipitation are the predominant extraction methods, and varying conditions in these methods may affect the yield and quality of SPI. In particular, the pH is one of the most important factors in alkaline extraction. Acidic amino acids tend to ionize in an alkaline environment; therefore, an extremely high pH may reduce the quality of SPI [17]. In addition, several other factors, such as the temperature, extraction time, and material-to-liquid ratio, may affect the yield and quality of SPI. In acid precipitation, the highest yield of precipitated proteins is reported to be achieved at a pH of 4.5 [18].

Soy protein products, especially SPI, have been widely used as food ingredients owing to their nutritional and functional properties [19]. The most commonly used form of SPI is soy protein powder [6]. SPI has excellent emulsifying, gelling, foaming, and film-forming properties and is considered a safe and stable food additive [20]. The water-holding capacity (WHC) of SPI directly determines the flavor, texture, and composition of the product and is closely associated with the preservation of freshness and shape during food storage [21]. The WHC of SPI is superior to that of protein extracted from quinoa [22]. SPI has a good oil-binding capacity (OBC), another important factor contributing to food’s flavor and texture; therefore, it can be added to meat products to stop fat loss and maintain shape [23]. In addition, SPI can be used to create fine and stable foams in cake batters and increase the fluffiness of bread. It has good foaming stability, which is beneficial to the expansion and stability of cakes [24,25,26]. Gel formation, a functional property of proteins, has received substantial attention in recent years owing to its role in maintaining the texture and sensorial perceptions of food products [27,28]. The structural matrix of gels is used to preserve water, flavor, sugar, and ingredients [29].

DND358 is a new hypocholesterolemic soybean cultivar lacking a subset of allergenic protein subunits, developed through a three-way cross [30]. We have previously shown that the cholesterol-lowering effects of soy protein powder prepared from defatted DND358 soy flour, are similar to those of fenofibrate, which is a lipid-lowering drug [31]. Based on these findings, this study aimed to optimize the extraction method of DND358-SPI, evaluate the processing adaptability of DND358-SPI, and provide a theoretical basis for the use of DND358-SPI in various types of processed foods.

## 2. Materials and Methods

### 2.1. Experimental Materials

The Dongnong47 (DN47), Dongnong42 (DN42), and Heihe43 (HH43) soybean varieties, which contain all 7S and 11S subunits, were used in this study. DN47 is a high-oil soybean variety, DN42 is a high-protein soybean variety, and HH43 is a universal soybean variety. In addition, Dongnongdou358 (DND358), which lacks 7S α subunits and G1, G2, and G4 11S subunits, was also used. These soybean cultivars were planted in a field at the Northeast Agricultural University Experimental Station (Harbin, China) during 2023. All experimental materials were provided by the Northeast Agricultural University.

### 2.2. Preparation of SPI

Soybeans were ground in a flour milling machine, and the flour was defatted two times. The defatted flour was dissolved in deionized water at a ratio of 1:10 (*w*/*v*) and shaken at 50 °C for 50 min, with the pH adjusted to 8.5. This suspension was centrifuged at 8000× *g* for 20 min at 4 °C, followed by the collection of the supernatant. The protein in the supernatant was precipitated at a pH of 4.5, and the resulting precipitate was resolubilized in deionized water at 50 °C and a pH of 7. The protein solution was stored at −80 °C for 24 h, followed by freeze-drying.

### 2.3. SDS-PAGE Analysis of SPI

A total of 0.01 g of freeze-dried SPI powder was dissolved in SDS sample buffer (comprising 2% SDS, 5% 2-mercaptoethanol, 10% glycerol, 5-M urea, and 62.5-mM Tris). After the solution was centrifuged at 15,000× *g*, 4.5% stacking and 12.5% separating polyacrylamide gels were used to separate the proteins in 10 µL of the supernatant. The separated proteins were stained with Coomassie Brilliant Blue R 250, and the gels were scanned using the SHARP JX-330 scanner (Amersham Biosciences, Baie d’Urfe, Quebec, Canada). The levels of each subunit of the 7S and 11S proteins were calculated as the percentage of the area of the subunit with respect to the total 7S or 11S area.

### 2.4. Isoflavone Analysis

HPLC was used to detect three isoflavones, namely daidzein, glycitein, and genistein, in SPIs derived from the four soybean cultivars. Briefly, a total of 0.25 g of soybean flour was dissolved in 10 mL of 80% HPLC-grade methanol in water. The solution was incubated in a water bath at 65 °C for 2 h with constant stirring. After the solution was cooled to room temperature, it was incubated with 0.3 mL of 2-M sodium hydroxide on a shaker at room temperature for 10 min. The solution was mixed with 0.1 mL of glacial acetic acid, and a final volume of 5 mL was obtained by adding methanol. Subsequently, 0.4 mL of water was added to 0.5 mL of the supernatant, and a final volume of 1 mL was obtained by adding methanol. After this solution was centrifuged at 1500 rpm for 10 min, the supernatant was passed through a 0.2 nm membrane filter and collected in fresh bottles for liquid chromatography. Each sample was analyzed in triplicate. For chromatography, the Inertsil ODS4 column (4.6 mm × 250 mm, 5 µm) was used at 40 °C, with mobile phase A consisting of water, methanol, and acetic acid (44:5:1, *v*/*v*/*v*) and mobile phase B consisting of methanol and acetic acid (49:1, *v*/*v*). Chromatography was performed with the following parameters: wavelength, 260 nm; flow rate, 1.0 mL/min; and sample volume, 10 µL.

### 2.5. Detection of Amino Acids

The content of 7 essential amino acids, namely Met, Val, Lys, Ile, Phe, Leu, and Thr, and 10 non-essential amino acids, namely Asp, Ser, Glu, Gly, Ala, Cys, Tyr, His, Arg, and Pro, in four different SPIs was evaluated. To estimate the protein content, the nitrogen content was evaluated and multiplied by a conversion factor of 6.25. The total amino acids were extracted from seed meal hydrolyzed in 6-M HCl for 22 h in sealed evacuated tubes in boiling water maintained at 110 °C. The amino acid compositions of the hydrolysates were determined using the L-8800 amino acid analyzer (Hitachi, Tokyo, Japan). To extract the free amino acids, 5.00 g of seed meal (soybean seeds were collected using the sample quartile method, fully dried, ground using a milling machine, filtered through a 0.25 mm sieve, and thoroughly mixed) was finely homogenized in 30 mL of sulfosalicylic acid (10 g/100 mL) and disrupted ultrasonically for 30 min. Subsequently, the sample was centrifuged at 5000× *g* for 5 min, and the supernatant was passed through a 22 µm GD/X sterile disposable syringe filter. The filtrate was analyzed using the L-8800 amino acid analyzer, and the concentration of amino acids was calculated as follows: g/16-g N in the test protein sample divided by g/16-g N in the scoring pattern.

### 2.6. Single-Factor Experiments and Orthogonal Test

Single-factor experiments were used to assess the effects of four factors (pH, temperature, time, and material–liquid ratio) on the extraction of DND358-SPI. The pH was set at 8.0, 8.5, 9.0, 9.5, and 10.0. The temperature was set at 40 °C, 45 °C, 50 °C, 55 °C, and 60 °C. The extraction time was set at 40 min, 50 min, 60 min, 70 min, and 80 min, and the material-to-liquid ratio was set at 1:10, 1:12.5, 1:15, 1:17.5, and 1:20.

The suitable ranges for the pH, temperature, time, and material-to-liquid ratio were obtained based on single-factor experiments. Subsequently, we designed a four-factor, three-level L9 (3^4^) orthogonal test to analyze the factors affecting the rate of SPI extraction from DND358. The extraction rate of the protein was used as an evaluation index to determine the highest extraction rate of DND358-SPI under different conditions. Three replicates were used for both experiments.

### 2.7. Water-Holding Capacity

The WHC was determined using the method described by Beuchat (1977) [32], with slight modifications. A total of 1 g of the sample was dissolved in 20 mL of water in a pre-weighed 50 mL centrifuge tube, and the solution was centrifuged at 10,000 rpm/min for 5 min to extract the supernatant. If no supernatant was extracted, more water was added and centrifugation was continued until a supernatant appeared. Subsequently, the sediment in the centrifuge tube was weighed after the removal of the supernatant, and the WHC was calculated using the following formula:WHC (%) = (m_1_ − m)/m(1)

In the abovementioned equation, m_1_ represents the weight of the sediment (g) and m represents the weight of the original sample (g).

### 2.8. Oil-Binding Capacity

The OBC was determined using the method described by Beuchat (1977) [32]. Briefly, a total of 2 g of the sample was mixed with 20 mL of soybean oil in a pre-weighed 50 mL centrifuge tube. After the mixture was centrifuged at 3000 r/min for 10 min at room temperature, the supernatant was removed and the residue was weighed. Subsequently, the OBC was calculated using the following formula:OBC (%) = (m_1_ − m)/m(2)

In the abovementioned equation, m_1_ represents the weight of the sediment (g) and m represents the weight of the original sample (g).

### 2.9. Emulsification Property

The emulsifying property of SPI was determined using the method described by Klompong et al. (2007) [33]. Briefly, 10 mL of soybean oil and 30 mL of 2% protein solution were mixed using a magnetic stirrer at 400 rpm/min for 2 min. The pH was adjusted to 7.0, and the solution was homogenized at 20,000 rpm/min for 1 min. Subsequently, 50 μL of liquid from the bottom of the solution was aliquoted and mixed with 5 mL of 0.1% SDS at 0 min and 10 min. The absorbance was measured at 500 nm. The emulsifying activity index (EAI) and emulsion stability index (ESI) were calculated using the following formulas [33]:(3)$$E\mathrm{AI}\;(m^{2}/g)=(2\;\times \;2.303\;\times \;A_{0})/F\;\times \;m$$
(4)$$\mathrm{ESI}\;(\%)=(A_{0}-A_{10})/A_{0}\;\times \;100$$

In the abovementioned equations, F represents the volume fraction of oil in the emulsion (0.25), m represents the weight of SPI (g), and A_10_ and A_0_ represent the absorbance of the protein emulsion at 500 nm at 10 min and 0 min, respectively.

### 2.10. Foaming Capacity and Stability

A total of 2 g of SPI was mixed with 100 mL of distilled water on a magnetic stirrer at 400 rpm/min for 2 min. The pH was adjusted to 7.0, and the solution was homogenized at 15,000 rpm/min for 2 min. The solution was immediately transferred to a 250 mL measuring cylinder and allowed to stand for 20 s. Subsequently, the foaming capacity (FC) was calculated using the following formula [33]:FC (%) = (V_2_ − V_1_)/V_1_ × 100(5)

In the abovementioned equation, V_1_ represents the volume before stirring (mL), whereas V_2_ represents the volume after stirring (mL).

The stirred samples were allowed to stand for 30 min at room temperature, and the foaming stability (FS) was calculated using the following formula:FS (%) = (V_3_ − V_1_)/V_1_ × 100(6)

In the abovementioned equation, V_1_ represents the volume before stirring (mL), whereas V_3_ represents the volume after standing (mL).

### 2.11. Gelation Ability

A total of 34 g of SPI was transferred to a beaker and mixed with 200 mL of distilled water. A glass rod was used to stir the mixture until the SPI was dissolved uniformly, followed by homogenization at a high speed for 2 min. The beaker was sealed with plastic wrap and incubated in a water bath at 100 °C for 10 min. After the beaker was removed and immediately cooled to room temperature, 0.06 mol/L gluconolactone was added and the solution was homogenized at 20,000 rpm/min for 40 s. The homogenized solution was poured into molds, which were sealed with plastic wrap and incubated in a water bath at 80 °C for 30 min. Subsequently, the molds were immediately placed in an ice bath to cool them to room temperature. The samples were slowly separated from the molds and transferred to trays, stored at −20 °C for 24 h, and analyzed for gelation using the CTX texture analyzer (Ametex Brookfield, Middleborough, MA, USA).

### 2.12. Statistical Analysis

All data were expressed as the mean ± standard deviation (SD). One-way analysis of variance (ANOVA) followed by the post hoc Tukey HSD test was performed to estimate the differences among the groups. The IBM SPSS Statistics software (version 22.0) was used for statistical analysis. A *p*-value of <0.05 was considered statistically significant.

## 3. Results and Discussion

### 3.1. Identification of Protein Subunit Composition of SPI from DND358

SPIs were obtained from four soybean cultivars, namely DND358, DN47, DN42, and HH43. The results of the SDS-PAGE of the SPIs are demonstrated in Figure 1. DN47-SPI, DN42-SPI, and HH43-SPI contained all 7S and 11S subunits (lanes 3, 4, 5, 6, 7, and 8), and their protein profiles were consistent with those of previously reported soybean varieties. In DND358-SPI (lanes 1 and 2), the absence of α subunits resulted in the compensatory accumulation of the α′ subunit of 7S, which was consistent with the findings of our previous study [31]. The enrichment of the α’ subunit is an environmentally independent genetic trait.

### 3.2. Characterization of DND358-SPI

Table 1 shows the comparison of the amino acid compositions of the SPI samples from the four soybean varieties. DND358-SPI had a high concentration of all essential amino acids. In particular, the concentrations of valine, lysine, phenylalanine, and threonine in DND358-SPI were significantly higher than those in the other varieties (3.65%, 5.39%, 4.38%, and 3.24%, respectively). For non-essential amino acids, DND358-SPI had the highest concentrations of glutamate, cysteine, tyrosine, histidine, and proline at 16.57%, 1.11%, 3.14%, 2.67%, and 4.42%, respectively. These results indicated that the four deletion mutations in DND358-SPI significantly affected the concentrations of amino acids in the SPIs.

The levels of isoflavones in the four SPIs are shown in Table 2. The concentrations of daidzein and glycitein were significantly higher in DND358-SPI than in the SPIs of the other three varieties, accounting for 28.59% and 9.22% of the total isoflavone content. HH43-SPI had the highest concentration of total isoflavones at 2803.13 μg/g, which may be attributed to its high genistein content. DND358-SPI had the highest protein content at 88.01% and the highest Phe content at 4.38% (Figure 2, Table 1). Phenylalanine metabolism is the most important pathway in soybean isoflavone synthesis. According to the results, a potential relationship was observed among the protein, isoflavone, and Phe content of DND358-SPI, suggesting that DND358-SPI had good physicochemical and functional properties.

### 3.3. Orthogonal Test on Optimal Extraction Process for DND358-SPI

The rate of protein extraction increased with an increase in the pH of alkaline extraction and reached the maximum value at a pH of 9.5 (Figure 3A). These results indicated that DND358-SPI had a high concentration of acidic amino acids, which tend to undergo ionization in an alkaline environment [17]. As the pH increased, the extraction rate decreased because the increased alkalinity caused the extreme denaturation of the proteins, with some proteins undergoing hydrolysis. Furthermore, the temperature affected the stability of the SPI [34]. Soybean proteins are denatured at a high temperature, with the denaturation temperature of glycinin being 92 °C and that of β-conglycinin being 71 °C [35,36]. The rate of protein extraction increased with an increase in the temperature of alkaline extraction and reached the maximum value at 55 °C (Figure 3B). As the temperature continued to increase, the extraction rate decreased owing to the denaturation of some proteins. The rate of protein extraction increased with an increase in the duration of alkaline extraction and reached the maximum value at a duration of 70 min (Figure 3C). However, the extraction rate decreased at 80 min, which may be attributed to the evaporation of the solvent over a prolonged period. The rate of protein extraction increased with an increase in the material-to-liquid ratio and reached the maximum value at a ratio of 1:15 (Figure 3D). However, it began to decrease when the feed-to-liquid ratio continued to increase further. Consequently, the material-to-liquid ratios of 1:12.5, 1:15, and 1:17.5 were selected for subsequent analysis.

The optimal conditions for DND358-SPI extraction are shown in Table 3. Based on the abovementioned results, we selected the following four factors for an orthogonal test: a pH of 9.0, 9.5, and 10.0; a temperature of 50 °C, 55 °C, and 60 °C; extraction time of 60 min, 70 min, and 80 min; and a material-to-liquid ratio of 1:15, 1:17.5, and 1:20. A four-factor, three-level L9 (3^4^) orthogonal test was performed to analyze the factors affecting the rate of protein extraction from DND358. As shown in Table 4, the order of the factors affecting the protein extraction rate is as follows: material-to-liquid ratio (D) > temperature (B) > time (C) > pH (A). According to the highest K value of each factor (A_2_B_2_C_3_D_3_), the optimal conditions for the alkaline extraction of DND358 were identified as follows: pH, 9.5; temperature, 55 °C; duration, 80 min; and material-to-liquid ratio, 1:20. The average extraction rate of DND358-SPI was 38.71%, which was the highest among all factor combinations.

### 3.4. Functional Properties of DND358-SPI

The WHCs of quinoa protein isolate (QPI) and wheat protein isolate are 3.94 ± 0.06 mL/g and 3.67 ± 0.05 mL/g, respectively [18]. In this study, the WHC of DND358-SPI was 4.73 ± 0.03 g/g, which suggests that DND358-SPI is suitable for use as a functional additive in bakery products to enhance their freshness and retain their shape (Table 5). The oil-binding capacity (OBC) of SPI was evaluated to assess its ability to reduce serum cholesterol levels [37]. The OBC of DND358-SPI (3.00%) was significantly higher than that of the SPIs derived from the other three soybean varieties. These results suggested that DND358-SPI had a good cholesterol-lowering ability.

The FC of DND358-SPI was estimated to be 14.67% in a 2% protein solution. The enzyme-mediated hydrolysis of SPI may improve foam formation because small hydrolyzed proteins are more likely to move to the gas–liquid interface [38]. SPIs have high foaming and emulsion stability, possibly owing to their enhanced steric repulsive forces in colloidal systems [39,40]. DN47-SPI had a particularly high FC at 22.33%, which highlights its potential as an ideal additive for sparkling beverages or pastries. DND358-SPI exhibited lower emulsion stability and emulsifying abilities than the SPIs derived from the other three soybean varieties. The emulsifying activity index (EAI) and emulsion stability index (ESI) are two important functional properties of SPIs that influence the selection of proteins for industrial processes [41].

The gelation ability of DND358-SPI was significantly stronger than that of the SPIs derived from the other three soybean varieties (Table 6). The hardness, gumminess, chewiness, and resilience of the gel formed by DND358-SPI (159.27 g, 186.07 g, 6.78 mj, and 1.88, respectively) were higher than those of the gels formed by the other three SPIs. The gelation of 7S globulins did not involve the -SH/S-S exchange reaction; therefore, the resulting gel was soft and clear. However, the gel formed from 11S globulins was hard and turbid [42]. With an increase in the 11S/7S ratio, more covalent bonds were generated through disulfide bonding, as more total cysteine groups are present in 11S proteins than in 7S proteins. These stronger molecular forces (covalent bonding) may increase the firmness of tofu, which is prepared from soy milk [43].

The hardness of a gel increases with an increase in the protein concentration [44]. DND358-SPI had the highest protein content and the highest hardness. High gumminess and chewiness can improve the quality of tofu [45]. In this study, the gumminess and chewiness of different SPIs were compared. Specifically, DND358-SPI had the highest gumminess and chewiness (Table 6), indicating that DND358 is suitable for preparing tofu.

## 4. Conclusions

In this study, we analyzed the amino acid compositions of SPIs derived from four soybean cultivars. In particular, DND358-SPI had a high concentration of all essential and non-essential amino acids. Subsequently, we focused on the extraction process and functional properties of DND358-SPI. By optimizing the conditions of alkaline extraction, the yield of DND358-SPI could be increased and cost-effective extraction could be achieved. The results indicate that DND358-SPI can be used as an additive in not only beverages but also in several other food products. Owing to its high WHC and OBC, DND358-SPI can be used as an additive in confectionery to maintain a soft texture. In addition, it can form hard gels, which may improve the texture and elasticity of tofu and maintain its stability. Therefore, DND358 can be used to realize specialization and traceability in many fields. This study provides a theoretical basis for the use of DND358 in different types of processed foods.

## Figures and Tables

**Figure 1 foods-13-03236-f001:**
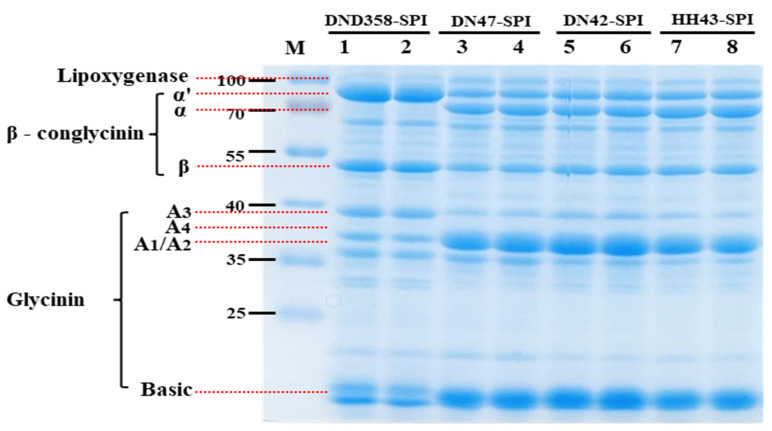
Results of SDS-PAGE of subunits of 7S and 11S globulins in four different SPIs. M, marker protein. Lanes 1 and 2: DND358-SPI (null α, G1, G2, and G4); Lanes 3 and 4: DN47-SPI (normal); Lanes 5 and 6: DN42-SPI (normal); Lanes 7 and 8: HH43-SPI (normal).

**Figure 2 foods-13-03236-f002:**
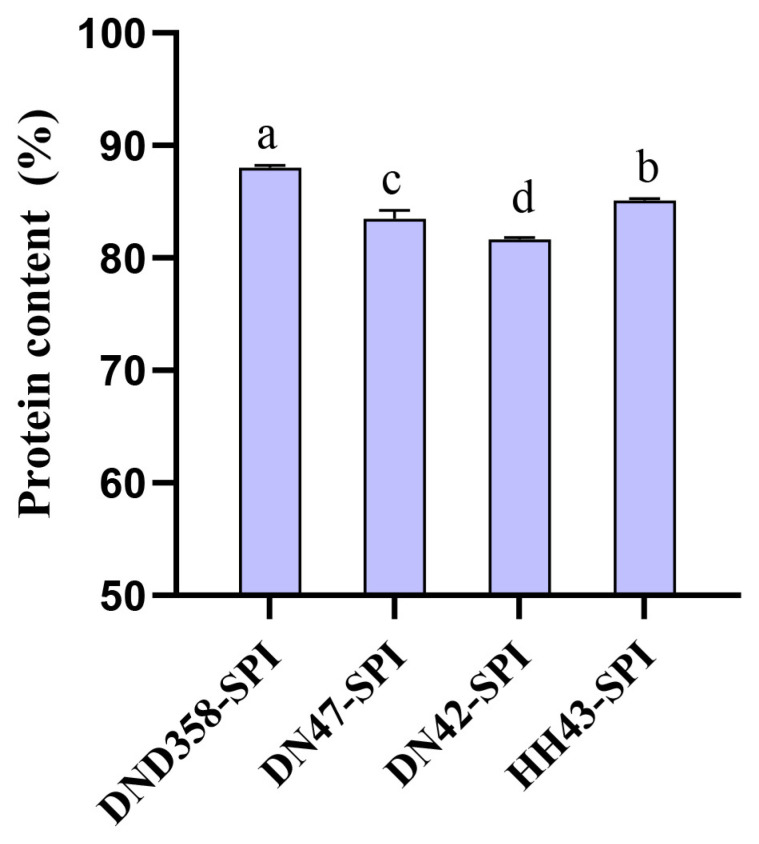
Analysis of total proteins in four SPIs. The letters on the bars indicate significant differences (*p* < 0.05).

**Figure 3 foods-13-03236-f003:**
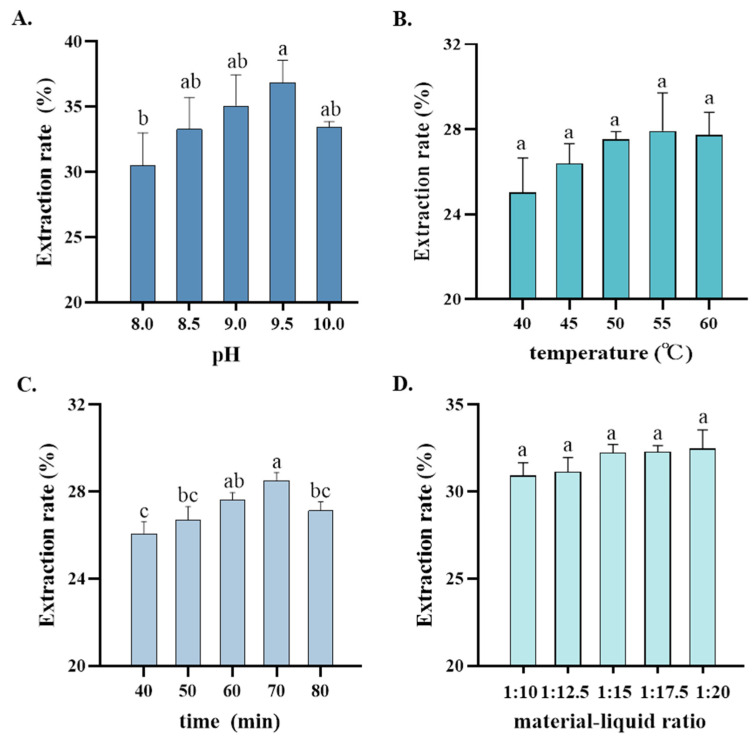
Single-factor experiments assessing optimal conditions for DND358-SPI extraction. (**A**–**D**) Effects of pH, temperature, time, and material-to-liquid ratio on the alkaline extraction of DND358-SPI. The different letters on the bars indicate significant differences at the 5% probability level as determined using the Tukey HSD test.

**Table 1 foods-13-03236-t001:** Amino acid compositions of SPIs obtained from mature seeds of four soybean varieties.

Amino Acid (%)	DND358-SPI	DN47-SPI	DN42-SPI	HH43-SPI
EAAs				
Met	1.17 ± 0.01 ^a^	1.20 ± 0.01 ^a^	1.08 ± 0.03 ^b^	0.98 ± 0.02 ^c^
Val	3.65 ± 0.02 ^a^	3.51 ± 0.03 ^b^	3.40 ± 0.04 ^c^	3.00 ± 0.03 ^d^
Lys	5.39 ± 0.03 ^a^	5.16 ± 0.06 ^b^	4.80 ± 0.09 ^c^	4.52 ± 0.08 ^d^
Ile	3.77 ± 0.01 ^b^	3.92 ± 0.05 ^a^	3.39 ± 0.08 ^c^	3.39 ± 0.05 ^c^
Phe	4.38 ± 0.01 ^a^	4.33 ± 0.03 ^b^	3.92 ± 0.04 ^b^	3.83 ± 0.07 ^b^
Leu	6.40 ± 0.01 ^b^	6.70 ± 0.05 ^a^	6.29 ± 0.12 ^b^	6.28 ± 0.10 ^b^
Thr	3.24 ± 0.01 ^a^	2.97 ± 0.03 ^b^	2.52 ± 0.05 ^d^	2.75 ± 0.05 ^c^
Total EAAs	28.01 ± 0.06 ^a^	27.80 ± 0.25 ^a^	25.40 ± 0.33 ^b^	24.74 ± 0.39 ^b^
Non-essential AAs				
Asp	9.74 ± 0.01 ^b^	9.64 ± 0.11 ^b^	10.23 ± 0.09 ^a^	8.04 ± 0.10 ^c^
Ser	4.35 ± 0.01 ^a^	4.35 ± 0.07 ^a^	3.52 ± 0.06 ^c^	3.99 ± 0.08 ^b^
Glu	16.57 ± 0.07 ^a^	16.42 ± 0.13 ^a^	15.75 ± 0.29 ^b^	14.98 ± 0.20 ^c^
Gly	3.39 ± 0.01 ^a^	3.35 ± 0.04 ^a^	3.39 ± 0.01 ^a^	2.90 ± 0.05 ^b^
Ala	3.04 ± 0.02 ^b^	3.41 ± 0.03 ^a^	2.80 ± 0.13 ^c^	2.99 ± 0.04 ^b^
Cys	1.11 ± 0.01 ^a^	1.00 ± 0.02 ^b^	0.79 ± 0.04 ^d^	0.85 ± 0.01 ^c^
Tyr	3.14 ± 0.01 ^a^	3.11 ± 0.02 ^a^	2.73 ± 0.05 ^b^	2.65 ± 0.04 ^b^
His	2.67 ± 0.01 ^a^	2.10 ± 0.02 ^b^	2.06 ± 0.03 ^b^	1.89 ± 0.04 ^c^
Arg	6.12 ± 0.02 ^ab^	6.18 ± 0.06 ^a^	5.79 ± 0.10 ^c^	5.98 ± 0.06 ^b^
Pro	4.42 ± 0.08 ^a^	4.18 ± 0.04 ^b^	3.10 ± 0.13 ^d^	3.91 ± 0.08 ^c^
Total AAs	82.57 ± 0.24 ^a^	81.54 ± 0.75 ^a^	75.55 ± 1.24 ^b^	72.92 ± 1.04 ^c^

AAs, amino acids; EAAs, essential amino acids. The different letters in the rows indicate significant differences at the 5% probability level as determined using the Tukey HSD test.

**Table 2 foods-13-03236-t002:** Analysis of isoflavone content in four SPIs.

Cultivar-SPI	Daidzein (µg/g)	Glycitein (µg/g)	Genistein (µg/g)	Total Isoflavones (µg/g)
DND358-SPI	787.84 ± 2.27 ^a^	254.14 ± 3.53 ^a^	1713.44 ± 1.60 ^c^	2755.41 ± 5.42 ^b^
DN47-SPI	669.58 ± 1.01 ^c^	116.63 ± 0.47 ^c^	1801.71 ± 0.71 ^b^	2587.91 ± 2.13 ^c^
DN42-SPI	647.97 ± 2.58 ^d^	103.48 ± 0.41 ^d^	1366.81 ± 1.34 ^d^	2118.25 ± 4.28 ^d^
HH43-SPI	759.68 ± 1.05 ^b^	145.64 ± 2.96 ^b^	1897.80 ± 1.86 ^a^	2803.13 ± 5.52 ^a^

The different letters in the columns indicate significant differences at the 5% probability level as determined using the Tukey HSD test.

**Table 3 foods-13-03236-t003:** Single-factor experiments assessing optimal conditions for DND358-SPI extraction.

Level	pH	Temperature (°C)	Time (min)	Material–Liquid Ratio
1	9	50	60	1:15
2	9.5	55	70	1:17.5
3	10	60	80	1:20

**Table 4 foods-13-03236-t004:** Orthogonal test results of DND358-SPI extraction process.

No.	Factor				Extraction Rate (%)
	A	B	C	D	
1	1	1	1	1	31.20
2	1	2	2	2	35.33
3	1	3	3	3	37.33
4	2	1	2	3	37.07
5	2	2	3	1	33.47
6	2	3	1	2	35.33
7	3	1	3	2	35.73
8	3	2	1	3	37.33
9	3	3	2	1	30.53
K_1_	103.86	104.00	103.86	95.20	
K_2_	105.87	106.13	102.93	106.39	
K_3_	103.59	103.19	106.53	111.73	
R_j_	2.28	2.94	3.60	16.53	

A: pH; B: temperature; C: time; D: material-to-liquid ratio.

**Table 5 foods-13-03236-t005:** Functional characteristics of four SPIs.

Sample	WHC (g/g)	OBC (g/g)	FC (%)	FS (%)	EAI (m^2^/g)	ESI (%)
DND358-SPI	4.73 ± 0.03 ^a^	3.00 ± 0.03 ^a^	14.67 ± 1.15 ^b^	61.61 ± 4.64 ^b^	10.67 ± 0.45 ^a^	18.37 ± 2.78 ^c^
DN47-SPI	2.11 ± 0.12 ^c^	1.98 ± 0.12 ^b^	22.33 ± 2.52 ^a^	74.58 ± 1.68 ^a^	10.89 ± 1.30 ^a^	35.44 ± 2.18 ^ab^
DN42-SPI	2.44 ± 0.10 ^b^	1.84 ± 0.14 ^b^	10.67 ± 1.15 ^b^	62.22 ± 3.85 ^b^	11.27 ± 1.32 ^a^	36.30 ± 2.62 ^a^
HH43-SPI	2.75 ± 0.18 ^b^	2.03 ± 0.11 ^b^	14.33 ± 1.53 ^b^	65.34 ± 3.49 ^ab^	11.13 ± 0.90 ^a^	30.20 ± 1.13 ^b^

WHC, water-holding capacity; OBC, oil-binding capacity; FC, foaming capacity; FS, foaming stability; EAI, emulsifying activity index; ESI, emulsion stability index. The different letters in the columns indicate significant differences at the 5% probability level as determined using the Tukey HSD test.

**Table 6 foods-13-03236-t006:** Gelation ability of four SPIs.

Sample	Hardness (g)	Springiness (mm)	Cohesiveness	Gumminess (g)	Chewiness (mj)	Resilience
DND358-SPI	159.27 ± 0.35 ^a^	3.70 ± 0.17 ^a^	1.22 ± 0.07 ^a^	186.07 ± 5.76 ^a^	6.78 ± 0.16 ^a^	1.88 ± 0.04 ^a^
DN47-SPI	121.60 ± 5.50 ^b^	4.06 ± 0.06 ^a^	1.10 ± 0.01 ^a^	177.50 ± 9.13 ^a^	4.90 ± 0.31 ^b^	1.62 ± 0.11 ^b^
DN42-SPI	49.57 ± 2.47 ^d^	3.86 ± 0.36 ^a^	1.10 ± 0.05 ^a^	53.93 ± 1.67 ^c^	2.19 ± 0.21 ^c^	1.30 ± 0.03 ^c^
HH43-SPI	101.73 ± 1.33 ^c^	3.90 ± 0.27 ^a^	1.15 ± 0.08 ^a^	110.53 ± 5.98 ^b^	4.38 ± 0.24 ^b^	1.35 ± 0.07 ^c^

The different letters in the columns indicate significant differences at the 5% probability level as determined using the Tukey HSD test.

## Data Availability

The original contributions presented in the study are included in the article, further inquiries can be directed to the corresponding author.

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
