# Peer review of "Physicochemical and Functional Properties of DND358 (A Hypocholesterolemic Soybean) Protein Isolate"

_foods, 2024, doi:10.3390/foods13203236_

Round 1
Reviewer 1 Report
Comments and Suggestions for Authors
Physicochemical and Functional Properties of DND358 2 (A Hypocholesterolemic Soybean) Protein Isolate
The work has a well-defined structure, presents scientific relevance, and the results are well described and discussed. As it involves optimization, it should include the ideal point as well as its validation.
Abstract
Comment 1:
Lines 15-16: “and to investigate its potential applications in the food industry.”
Lines 22-23: “These findings provided a theoretical basis for the wide-ranging applica-22 tions of DND358-SPI in the food industry.”
What are the potential applications of DND358-SPI in the food industry? The protein isolate that was extracted in the study in question can be used for which applications? It would be interesting to add this information to the abstract.
Keywords
Comment 2: Do not use keywords that are described in the title (DND358, functional properties)
Introduction
Comment 3: Line 40: “Soy protein isolate (SPI) is a complete protein with an ideal amino acid profile.”
It may be interesting to add the main amino acids present in soy protein to the introduction.
Comment 4: Lines 71-72: What are the allergenic proteins present in the DND358 soybean cultivar?
Materials and Methods
Comment 5: Lines 80-84: Where were the samples grown? In what year were they grown?
Comment 6: Lines 80-84: After the harvest of the grains, how was the pre-industrialization carried out? Were these samples dried and stored until they were used in the study? The post-harvest stages can influence the properties of the protein isolate.
Comment 7: Line 101-114: Add which isoflavones were quantified.
Comment 8: Lines 115-128: Add which amino acids were quantified.
Results and Discussion
Comment 9: Line 216: DND358-SPI had a high concentration of all essential amino acids.
Were the location and year of cultivation the same? As well as the management practices during the cultivation of these genotypes? Soil conditions, climate, and management of fertilization and pest and disease control can influence the amino acid and isoflavone profile of soybean grains.
Comment 10: Lines 281-282: This result suggested that DND358-SPI had good cholesterol lowering ability
For a future study, it would be interesting to test DND358-SPI protein isolates in vivo to observe their actual impact on cholesterol levels.
Author Response
For research article
|
Response to Reviewer 1 Comments |
|
Thank you very much for taking the time to review this manuscript. Please find the detailed responses below and the corresponding corrections highlighted in the re-submitted files
|
|
Point-by-point response to Comments and Suggestions for Authors |
|
Comments 1: Abstract Lines 15-16: “and to investigate its potential applications in the food industry.” Lines 22-23: “These findings provided a theoretical basis for the wide-ranging applications of DND358-SPI in the food industry.” What are the potential applications of DND358-SPI in the food industry? The protein isolate that was extracted in the study in question can be used for which applications? It would be interesting to add this information to the abstract. |
|
Response: Previous studies have demonstrated that the cholesterol-lowering effects of DND358-SPI are similar to those of fenofibrate. Therefore, we speculated that DND358-SPI might be used to produce cholesterol-lowering food products. We have revised the abstract accordingly; please refer to lines 21–22. |
|
Comments 2: Keywords Do not use keywords that are described in the title (DND358, functional properties) |
|
Response: Thank you for your suggestion. We have revised the keywords accordingly; please refer to line 23. |
|
Comments 3: Introduction Line 40: “Soy protein isolate (SPI) is a complete protein with an ideal amino acid profile.” It may be interesting to add the main amino acids present in soy protein to the introduction. |
|
Response: Thank you for your suggestion. We have added the amino acid-related information to the Introduction section; please refer to lines 40–41. |
|
Comment 4: Lines 71-72: What are the allergenic proteins present in the DND358 soybean cultivar? |
|
Response: 7S and 11S globulins are the allergenic proteins in soybeans. DND358 lacks the α subunit of 7S globulin and the G1, G2, and G4 subunits of 11S globulin; therefore, DND358 is a soybean cultivar lacking a subset of allergenic protein subunits. |
|
Comment 5: Materials and Methods Lines 80-84: Where were the samples grown? In what year were they grown? |
|
Response: We have added the cultivation time and location of DND358 in the revised article; please refer to lines 85–87. |
|
Comment 6: Lines 80-84: After the harvest of the grains, how was the pre-industrialization carried out? Were these samples dried and stored until they were used in the study? The post-harvest stages can influence the properties of the protein isolate. |
|
Response: After harvest, the soybean seeds were stored in a dry condition until they were used in the study. |
|
Comment 7: Line 101-114: Add which isoflavones were quantified. |
|
Response: We have added the isoflavone types; please refer to lines 107–108. |
|
Comment 8: Lines 115-128: Add which amino acids were quantified. |
|
Response: We have specified the amino acids; please refer to lines 124–126. |
|
Comment 9: Results and Discussion Line 216: DND358-SPI had a high concentration of all essential amino acids. Were the location and year of cultivation the same? As well as the management practices during the cultivation of these genotypes? Soil conditions, climate, and management of fertilization and pest and disease control can influence the amino acid and isoflavone profile of soybean grains. |
|
Response: The four soybean cultivars were planted in the same location and year. The management practices, soil conditions, climate, and other conditions were the same. Therefore, there were no environmental or time differences. |
|
Comment 10: Lines 281-282: This result suggested that DND358-SPI had good cholesterol lowering ability For a future study, it would be interesting to test DND358-SPI protein isolates in vivo to observe their actual impact on cholesterol levels. |
|
Response: Thank you for your valuable feedback. In our future study, we will investigate DND358-SPI in vivo to assess its actual cholesterol-lowering effects. |
Reviewer 2 Report
Comments and Suggestions for Authors
The manuscript entitled “Physicochemical and Functional Properties of DND358 (A Hypocholesterolemic Soybean) Protein Isolate” presents interesting results regarding the preparation and characterization of DND358 SPI. The results are clearly presented, but in some places, they are too poorly discussed. I show some minor comments and suggestions below.
Below are the linguistic, editorial and substantive issues that should be taken into account:
Keywords
there is a lack of general research material in keywords i.e. soybean protein isolate
what do the authors mean? in my opinion too general
Introduction
Lines 44
Via - Latin phrases should be italicized
Line 62
BWH - ??
Line 66
Better than what?
From a linguistic point of view, Experimental Materials are not well formulated. I suggest rephrasing it.
Line 92
Free-dried? do you mean freeze-drying or air-drying?
Line 126
there is no need to provide the device manufacturer again if this has been done before
Table 1
The table already shows the results of the research carried out and should be in the Results section.
Lines 145-150; line 162
Transfer a total of 1 g of sample to a pre-weighed 50 mL centrifuge tube. Add 20 mL of water to dissolve it, followed by centrifugation at 10,000 rpm/min for 5 minutes to extract the supernatant. - this looks like an instruction; not a description of how it was done
Before sending the pdf, the authors should check it carefully; first of all, all equations are poorly formatted, and additionally, there is one comment in the work that changes the reading window and makes the review difficult.
Line 151
Sedimentheavy - ??
Line 160
Emulsifying properties or Emulsifying Activity Index (EAI) and Emulsion Stability Index (ESI)
EAI and ESI abbreviations should be explained in the headline or text
Line 206
missing citation
Line 209
“was or is” an environmentally independent genetic trait?
Figure 1
What does the term "normal" mean to authors?
Table 2
I suggest not to use abbreviations in the table description (A.A.)
Lines 223-228
this text should be under table 2; it will increase the readability and clarity of the work
Figure 2
the graph is too small
Line 227
…whereas DND358-SPI had the highest protein content at 88.01% (Figure 2).
Why is protein content discussed here and why is this information highlighted?
Section 3.3
definitely too little discussion with the literature; no references to the works of other authors; what is the denaturation temperature of soy protein? do other authors describe the influence of environmental factors?
Line 251
I wouldn't say it's growing rapidly (see statistical analysis)
Figure 3
the caption can be shortened (without repeating the same words for each of the A-D drawings)
Section 3.4
some shortcuts are introduced again, others are not; please be consistent in this regard
Line 287
DND358-SPI exhibited lower emulsion stability and emulsifying ability – lower than what?
Tables 5 and 6
please expand the caption above the table; the data does not concern only one variety but all four
Section 3.4
- poor discussion of the results of emulsifying and foaming properties; there should be a more extensive discussion in this area and more references to research by other scientists
- mixing food and cosmetic products (line 290 and sunscreens) why?
Line 279
table 5 not 6
Table 5
caption under the table; first explanation of abbreviations, then information about statistics (analogously as under other tables)
Line 316
Not only functional properties and extraction were studied
Line 318-319
We have generated a specialized potent variety of soybean named DND358, which has cholesterol-lowering effects. - the sentence should be rephrased; this aspect was not the subject of this research
Conclusions focus more on potential applications than on obtained results; more emphasis should be placed on characterization (including composition)
Comments on the Quality of English LanguageMinor editing of English language required.
Author Response
For review article
|
Response to Reviewer 2 Comments
|
|
Thank you very much for taking the time to review this manuscript. Please find the detailed responses below and the corresponding corrections highlighted in the re-submitted files.
|
|
Point-by-point response to Comments and Suggestions for Authors |
|
Keywords there is a lack of general research material in keywords i.e. soybean protein isolate what do the authors mean? in my opinion too general |
|
Response: We agree with this comment. We have revised the keywords; please refer to line 23. |
|
Introduction Lines 44 Via - Latin phrases should be italicized Line 62 BWH - ?? Line 66 Better than what? |
|
Response: Thank you for pointing this out. We have italicized via and other Latin phrases in the revised manuscript. We've adjusted this sentence; please refer to line 63. We have replaced “better” with “good”. |
|
From a linguistic point of view, Experimental Materials are not well formulated. I suggest rephrasing it. |
|
Response: |
|
We have rephrased this section accordingly. |
|
Line 92 Free-dried? do you mean freeze-drying or air-drying? |
|
Response: We mean freeze-drying; please refer to line 95. |
|
Line 126 there is no need to provide the device manufacturer again if this has been done before |
|
Response: Thank you for pointing this out. We have removed the device manufacturer details. |
|
Table 1 The table already shows the results of the research carried out and should be in the Results section. |
|
Response: Thank you for your suggestion. We have moved Table 1 to the Results and Discussion section. |
|
Lines 145-150; line 162 Transfer a total of 1 g of sample to a pre-weighed 50 mL centrifuge tube. Add 20 mL of water to dissolve it, followed by centrifugation at 10,000 rpm/min for 5 minutes to extract the supernatant. - this looks like an instruction; not a description of how it was done |
|
Response: We have revised this part; please refer to lines 153–155. |
|
Before sending the pdf, the authors should check it carefully; first of all, all equations are poorly formatted, and additionally, there is one comment in the work that changes the reading window and makes the review difficult. |
|
Response: Thank you for your valuable feedback. We have modified the formats of all equations in the revised manuscript. The comment has also been modified. |
|
Line 151 Sedimentheavy - ?? |
|
Response: The phrase has been revised to “the weight of the sediment”. |
|
Line 160 Emulsifying properties or Emulsifying Activity Index (EAI) and Emulsion Stability Index (ESI) EAI and ESI abbreviations should be explained in the headline or text |
|
Response: We have added the full forms of the abbreviations EAI and ESI in the revised manuscript. |
|
Line 206 missing citation |
|
Response: We have added the citation; please refer to line 222. |
|
Line 209 “was or is” an environmentally independent genetic trait? |
|
Response: Thank you for pointing this out. We have replaced “was” with “is”. |
|
Figure 1 What does the term "normal" mean to authors? |
|
Response: Normal signifies the presence of all 7S and 11S subunits in the SPI. |
|
Table 2 I suggest not to use abbreviations in the table description (A.A.) |
|
Response: We have used the full form of A.A. in the table description in the revised manuscript. |
|
Lines 223-228 this text should be under table 2; it will increase the readability and clarity of the work |
|
Response: Thank you for your suggestion. We have moved the text under Table 2 in the revised manuscript. |
|
Figure 2 the graph is too small |
|
Response: We have adjusted the figure to a suitable size. |
|
Line 227 …whereas DND358-SPI had the highest protein content at 88.01% (Figure 2). Why is protein content discussed here and why is this information highlighted? |
|
Response: DND358-SPI had the highest protein content at 88.01% and highest Phe content at 4.38%. Phenylalanine metabolism is the most important pathway in soybean isoflavone synthesis. Based on the results, we inferred a potential relationship among the protein, isoflavone, and Phe contents of DND358-SPI. These results suggest that DND358-SPI has good physicochemical and functional properties. Please refer to lines 245-250. |
|
Section 3.3 definitely too little discussion with the literature; no references to the works of other authors; what is the denaturation temperature of soy protein? do other authors describe the influence of environmental factors? |
|
Response: We have mentioned some reference articles in the Results and Discussion section of the revised manuscript. The denaturation temperature of SPI ranges from 71℃ to 92℃. Please refer to lines 263–265. |
|
Line 251 I wouldn't say it's growing rapidly (see statistical analysis) |
|
Response: We have deleted the word “rapidly”. |
|
Figure 3 the caption can be shortened (without repeating the same words for each of the A-D drawings) |
|
Response: We have deleted the repetitive words in the captions of Figures 3A–3D and mentioned the different conditions together. |
|
Section 3.4 some shortcuts are introduced again, others are not; please be consistent in this regard |
|
Response: We have modified the shortcuts and kept the format of citations consistent throughout the revised manuscript. |
|
Line 287 DND358-SPI exhibited lower emulsion stability and emulsifying ability – lower than what? |
|
Response: DND358-SPI exhibited lower emulsion stability and emulsifying ability than the SPIs derived from the other three soybean cultivars. Please refer to lines 312–313. |
|
Tables 5 and 6 please expand the caption above the table; the data does not concern only one variety but all four |
|
Response: We have expanded the caption above the table to mention all four soybean cultivars. |
|
Section 3.4 poor discussion of the results of emulsifying and foaming properties; there should be a more extensive discussion in this area and more references to research by other scientists mixing food and cosmetic products (line 290 and sunscreens) why? |
|
Response: We have mentioned some reference articles in this section in the revised manuscript. Please refer to lines 308–310. We have deleted the content mentioned in line 290. |
|
Line 279 table 5 not 6 |
|
Response: We have changed Table 6 to Table 5. |
|
Table 5 caption under the table; first explanation of abbreviations, then information about statistics (analogously as under other tables) |
|
Response: We have adjusted the order accordingly. |
|
Line 316 Not only functional properties and extraction were studied |
|
Response: We agree with this comment. We have mentioned other results in this part in the revised manuscript. |
|
Line 318-319 We have generated a specialized potent variety of soybean named DND358, which has cholesterol-lowering effects. - the sentence should be rephrased; this aspect was not the subject of this research |
|
Response: We agree with this comment. We have deleted this sentence. |
|
Conclusions focus more on potential applications than on obtained results; more emphasis should be placed on characterization (including composition) |
|
Response: We have added more information on the characterization of DND 358-SPI in this section. |
|
Minor editing of English language required. |
|
Response: Thank you for pointing this out. We will address this concern. |
Round 2
Reviewer 1 Report
Comments and Suggestions for Authors
The authors addressed all the questions and significantly improved the paper after the first review.